# Depletion of HIF-1α by Inducible Cre/loxP Increases the Sensitivity of Cultured Murine Hepatocytes to Ionizing Radiation in Hypoxia

**DOI:** 10.3390/cells11101671

**Published:** 2022-05-18

**Authors:** Akram Hamidi, Alexandra Wolf, Rositsa Dueva, Melanie Kaufmann, Kirsten Göpelt, George Iliakis, Eric Metzen

**Affiliations:** 1Institute of Physiology, Faculty of Medicine, University of Duisburg-Essen, Hufelandstraße 55, D45147 Essen, Germany; akram.hamidi@uk-essen.de (A.H.); alexandra_wolf_@gmx.net (A.W.); rositsa.dueva@uni-due.de (R.D.); melanie.kaufmann@uk-essen.de (M.K.); kirsten.goepelt@uni-due.de (K.G.); 2Institute of Medical Radiation Biology, Faculty of Medicine, University Duisburg-Essen, Hufelandstraße 55, D45147 Essen, Germany; george.iliakis@uk-essen.de; 3Institute of Physiology, University of Duisburg-Essen, Hufelandstraße 55, D45147 Essen, Germany

**Keywords:** hypoxia, HIF-1α, ionizing radiation, apoptosis, DNA damage repair

## Abstract

The transcription factor hypoxia-inducible factor (HIF) is the main oxygen sensor which regulates adaptation to cellular hypoxia. The aim of this study was to establish cultured murine hepatocyte derived cells (mHDC) as an in vitro model and to analyze the role of HIF-1α in apoptosis induction, DNA damage repair and sensitivity to ionizing radiation (IR). We have crossed C57/BL6 mice that bear loxP sites flanking exon 2 of Hif1a with mice which carry tamoxifen-inducible global Cre expression. From the offspring, we have established transduced hepatocyte cultures which are permanently HIF-1α deficient after tamoxifen treatment. We demonstrated that the cells produce albumin, acetylcholine esterase, and the cytokeratins 8 and 18 which functionally characterizes them as hepatocytes. In moderate hypoxia, HIF-1α deficiency increased IR-induced apoptosis and significantly reduced the surviving fraction of mHDC as compared to HIF-1α expressing cells in colony formation assays. Furthermore, HIF-1α knockout cells displayed increased IR-induced DNA damage as demonstrated by increased generation and persistence of γH2AX foci. HIF-1α deficient cells showed delayed DNA repair after IR in hypoxia in neutral comet assays which may indicate that non-homologous end joining (NHEJ) repair capacity was affected. Overall, our data suggest that HIF-1α inactivation increases radiation sensitivity of mHDC cells.

## 1. Introduction

In the mammalian organism, the liver plays a central role in protein, carbohydrate, and lipid metabolism. It is also the source of blood cells in fetal life, provides the blood with the vast majority of plasma proteins, produces bile, stores vitamins, and is vital for degradation and excretion of many xenobiotic substances. Given the great importance of the liver for all these processes, it is not surprising that transplantation is the only option in the event of irreversible acute or chronic liver failure. The liver is composed of a number of different cell types, namely hepatocytes, hepatic stellate cells (HSC), Kupffer cells, epithelial cells of bile ducts, and all cells of blood vessels. Virtually all the metabolic achievements of the liver depend on hepatocytes. HSC store lipid droplets and deposit connective tissue material in liver fibrosis, while Kupffer cells represent liver specific macrophages located in the liver sinusoids [1]. Primary human hepatocytes are scarcely available for biomedical research, even if protocols exist for short term culture of human hepatocytes [2] or have been developed for long term culture of hepatocyte-like cells differentiated from non-parenchymal epithelial cells [3]. Rodent hepatocytes have also been cultured [4], but apparently, greatly reduced complexity of the cell culture as compared to the in vivo situation and the highly proliferative state necessary for cell culture may lead to rapid de-differentiation of the cells. In particular, with respect to the vast number of genetically modified mice which have become commercially available, it would be highly desirable to have established a reproducible, long-term culture method of differentiated hepatocytes. In addition, hepatocyte culture technology would be invaluable in reduction and refinement of animal experimentation. 

The transcription factor complex hypoxia-inducible factor (HIF) is the main oxygen sensor which regulates adaptation to cellular hypoxia. HIF is composed of a β-subunit permanently present in the cell nucleus and an oxygen-regulated α-subunit [5]. Three distinct α-subunits have been characterized: HIF-1α is the best studied isoform and expressed ubiquitously, while HIF-2α shows more tissue-specific expression [6]. The significance of HIF-3α is largely unclear, although one splice variant of this gene locus has been reported to be a negative regulator of HIF-signaling [7]. In normoxia, proteasomal degradation of the α-subunit is triggered by prolyl hydroxylase domain protein (PHD) and oxygen dependent hydroxylation, followed by binding of von-Hippel-Lindau protein (pVHL) and subsequent ubiquitination [8], which eventually leads to proteasomal degradation. Under hypoxic conditions, HIF-1α is stable and undergoes translocation into the nucleus to dimerize with the constitutively expressed nuclear protein HIF-1ß/ARNT (aryl hydrocarbon receptor nuclear translocator) [9]. This dimer binds to hypoxia-responsive elements (HRE) on regulatory DNA regions and induces the transcription of HIF-1 target genes involved in glucose metabolism, angiogenesis, cell survival, erythropoiesis, and many other processes [10]. Given the complexity of hepatocyte metabolism and the multitude of HIF target genes, it is unsurprising that HIF in addition to its general effects on cell metabolism has been implicated in specific functions of hepatocytes: HIF-1α expressed in mouse hepatocytes is involved in obesity-induced glucose intolerance [11], in manganese excretion [12], and in the regulation of iron metabolism [13], to name only a few.

Of note, a number of processes of high biomedical importance are causative for chronic liver damage. For example, chronic hepatitis B and C virus infections as well as excessive ethanol consumption lead to a permanent attempt of hepatocyte regeneration. However, chronic inflammation and toxin exposure often exceed regeneration capacity and result in liver cirrhosis which predisposes patients to the development of hepatocellular carcinoma (HCC). Approximately half a million people worldwide are diagnosed with HCC, which is the fifth most common cancer among men and the seventh most common cancer among women. In addition, it is the third most common cause of cancer-related deaths worldwide [14]. In HCC, as in many other solid tumors, an inadequately low level of oxygen (hypoxia) is a hallmark of malignity [15] and arises mainly from the abnormalities in tumor vasculature. Hypoxia induces the activation of stress response pathways in tumor cells [16]. Some tumor cells change their gene expression profile to a more aggressive phenotype in response to hypoxia [17]. Importantly, HIF stabilization has been associated with tumor progression, metastasis, and reduced sensitivity to chemotherapy and radiation treatment [18,19,20] and is thus regarded as a marker predictive for poor patient outcome [21]. 

Approximately 50% of all tumor patients receive radiation therapy with curative or adjuvant intention. In patients with advanced HCC, very recent studies have demonstrated that stereotactic body radiotherapy (SBRT) is an efficient and safe treatment in HCC patients [22]. In general, ionizing radiation leads to DNA double strand breaks (DSB)—the most dramatic DNA damage that can occur in cells [23]. Irradiated cells heavily rely on four major DSB repair pathways: ”homologous recombination“ (HR), ”classical nonhomologous endjoining“ (c-NHEJ) [24], alternative end-joining (alt-EJ), and single strand annealing (SSA). At least HR and c-NHEJ can be affected by hypoxia [17]. 

The primary aim of this study was to establish long-term cultures of transduced murine hepatocyte derived cells (mHDC) as an in vitro model system. Specifically, we have generated mHDC with permanent and complete HIF-1α deficiency from genetically modified mice that allow tamoxifen-inducible genetic inactivation of HIF-1α. We have used this model to assess the function of HIF-1α in hepatocytes with respect to proliferation and apoptosis induction. In addition, it was particularly important to determine how HIF-1α signaling affects DNA repair.

## 2. Materials and Methods

### 2.1. Mice, Cell Culture, and Tamoxifen Administration

In co-operation with upcyte technologies GmbH (Hamburg, Germany), we aimed to establish cell cultures of differentiated murine hepatocytes with a deletion of HIF-1α. To this end, C57/BL6 mice with a global inducible HIF-1α knockout (KO) were generated by breeding C57BL/6-*Gt(ROSA)26Sor^tm9(Cre/ESR1)Arte^* (Model 10471, Taconic Biosciences, Lille Skensved, DK, USA) [25] to mice carrying loxP sites flanking exon 2 of HIF-1α [26]. Hepatocyte isolation, lentiviral transduction, and establishment of cell culture were performed by upcyte technologies on a commercial basis. Hepatocytes were isolated from three Cre negative mice (mouse 17, 23, and 24) which served as controls and three heterozygous ROSA_CreERT2 mice (mouse 16, 21, and 22). The hepatocytes were released from liver using a two-step collagenase perfusion technique. The resulting cells were counted and seeded in a subconfluent manner to allow initial growth in collagen coated cell culture vessels. The hepatocytes were transduced with lentiviral particles carrying the papilloma virus genes E6 and E7, which in turn induced expression of the oncostatin M receptor [27]. The lentiviral particles were used at 1 or 2 multiplicity of infection (MOI) 24 h after the isolation. The resulting upcyte^®^ hepatocytes are not fully immortalized. They maintain their typical cobblestone morphology and form a confluent monolayer before and after transduction. After the initial passages, upcyte^®^ mouse hepatocytes were transferred to our laboratory, gently thawed in the recommended thawing medium, and seeded at a density of 10,000 cells/cm^2^ in collagen-type I-coated flasks in Hepatocyte Culture Medium (HCM) containing supplement A, supplement B adapted for mouse (both from upcyte), 2 mmol/L L-glutamine (upcyte), 100 U/mL penicillin, 100 μg/mL streptomycin, and 5% fetal bovine serum (FBS). The medium was exchanged every 2–3 days. After reaching confluence, cells were subcultured with 0.25% trypsin/0.02% EDTA, and reseeded. To stop proliferation, the hepatocytes were cultivated in High Performance Medium (HPM = HCM without supplement B) for 3 days before endpoint experiments. Culture Medium (HCM), High Performance Medium (HPM), and Thawing Medium were all obtained from upcyte technologies. The cells were routinely cultured at 37 °C with 5% CO_2_ and 95% humidified air in a cell culture incubator. These conditions were also used for normoxic incubation of control cells. Hypoxia treatments were performed in a hypoxic chamber with 1% of O_2_, 5% CO_2_ and balance N_2_ (Toepffer Lab System, Goeppingen, Germany, or in an InVivo400, Baker Ruskinn, Bridgend, UK). Cre expression and the consecutive gene KO were induced by adding 500 nM 4H-tamoxifen to the hepatocyte cultures for three days. Tamoxifen was then removed by replacement of the HCM and not used at any later time-point.

### 2.2. Antibodies and Reagents

Antibodies against mouse keratin 8 and keratin 18 were from (Progen Biotechnik GmbH, Heidelberg, Germany) and used in a dilution 1:50 and 1:250 for immunofluorescent staining (IF). Anti-murine HIF-1α (1:1000 dilution for WB; Cayman Chemical Company, Ann Arbor, MI, USA), anti-PARP-1 (1:1000 dilution for WB; Cell Signaling Technologies, Danvers, MA, USA), anti-phospho-histone H2AX (1:1000 dilution for WB and 1:800 for IF; Cell Signaling Technologies), and anti-beta actin (1:6000 dilution; Abcam, Cambridge, UK). Horseradish peroxidase-coupled secondary antibodies used for WB, goat polyclonal anti-mouse (P0447), and anti-rabbit (P0448) were purchased from Dako (Santa Clara, CA, USA). Secondary antibodies used for IF were goat anti-rabbit IgG Alexa 488 (A11070), goat anti-mouse IgG Alexa 488 (A11017), all purchased from Invitrogen (Carlsbad, CA, USA). 

### 2.3. SDS PAGE and Western Blot

For all Western blots, whole cell lysates were used. Samples were lysed in RIPA buffer (50 mM Tris pH 7.5, 2 mM EDTA, 150 mM NaCl, 1% Nonidet P40, 0.1% SDS, 0.5% sodium desoxycholate and protease/phosphatase inhibitor cocktail (#5872, Cell Signaling)). All samples were prepared for electrophoresis in SDS-sample buffer (62.5 mM Tris (pH 7.4)), 3% β-mercaptoethanol, 2% SDS, 0.25 mg/mL bromophenol blue, 10% glycerol, 25 mM DTT). The sample proteins were separated by SDS-PAGE using 5–12.5% polyacrylamid gels and were transferred to a PVDF membrane using the Trans-Blot Turbo Blotting System (Bio Rad, CA, USA). The membranes were blocked with 5% BSA or 5% skimmed milk in TBS-T (50 mM Tris/HCl, 150 mM NaCl, 0.5% Tween-20, pH 7.2) for 1 h at room temperature and then incubated with the primary antibody overnight. Secondary HRP-conjugated antibodies were diluted in 5% skimmed milk in TBS-T and incubated with the membrane for 1 h the next day. Proteins were detected with the ECL Kit (ThermoFisher Scientific, Oberhausen, Germany) using a Fusion FX7 chemoluminescence documentation system (Peqlab/VWR International, Erlangen, Germany). 

### 2.4. Cell Viability and Colony Formation Assays

To determine cell viability, MTT assays were performed essentially as described previously [28]. In brief, 2 × 10^3^ cells were plated into 96-well plates, subjected to the intended treatment, and incubated for 72 h. The cells were then incubated with 3-(4,5-dimethylthiazol-2-yl)-2,5-diphenyltetrazolium bromide (MTT) for 4 h at 37 °C. Then the cells were lysed, and the absorbance of formazan was measured at 540 nm by photometry (Synergy HT, BioTek, Bad Friedrichshall, Germany). Colony formation assays were used to analyze cell survival after IR as given in a previous publication [29]. Briefly, the mHDC were seeded in rat collagen type I coated 6-well dishes. The cells were placed in hypoxia or normoxia for 48 h and irradiated. After 10 days the cells were fixed using 0.25% paraformaldehyde (PFA) for 20 min. The cells were stained with Coomassie brilliant blue (0.1 Coomassie brilliant blue, 5% acetic acid, 45% methanol). Colonies of more than 50 cells were counted. The plating efficiency (PE) was determined in control cells as the number of counted colonies/seeded cells. The survival fraction (SF) was calculated as the number of colonies formed after treatment/(cells seeded × PE).

### 2.5. Quantitative PCR

Following incubation in normoxia or hypoxia, total RNA was extracted using RNeasy^®^ Mini Kit (Qiagen, Hilden, Germany), according to the manufacturer’s protocol. Reverse transcription of the samples was performed using a QuantiTect Reverse Transcription Kit (Qiagen). Two micorgrams RNA were used for cDNA synthesis. Samples of cDNA were stored at −20 °C. Real-time polymerase chain reaction (qPCR) was performed as recommended by the manufacturer of MESA Green qPCRTM Mastermix Plus for SYBR Assay kits (Eurogentec, Seraing, Belgium). The oligonucleotide sequences used for amplification of Bcl2/adenovirus E1B 19 kDa-interacting protein 3 (BNIP3), glucose transporter 1 (GLUT1), phosphofructokinase (PFK), and nitric oxide synthase 3 (NOS3) cDNA are available on request. 

### 2.6. Radiation

Irradiation was performed with an X-ray machine (X-RAD 320, Precision X-ray, North Branford, CT, USA) operated at 320 kV, 12.5 mA with a 1.65 mm aluminum filter at a distance of 50 cm.

### 2.7. Immunofluorescence Staining 

Cytokeratin 8/18 staining was performed using approximately 80% confluent hepatocyte cultures. The cells were fixed with 1% paraformaldehyde (PFA) in PBS for 10 min and subsequently stained with the first antibodies diluted in 2% BSA for 1.5 h and then incubated in a 1:500 dilution of the Alexa Fluor 488 conjugated secondary antibody. The cells were counterstained with 1.5 µM Hoechst 33342 (Sigma-Aldrich, Munich, Germany) and placed on coverslips using mounting medium (Dako). Fluorescent images were captured using a Zeiss LSM 510 confocal microscope with a 63×/1.2 oil immersion lens (Carl Zeiss, Oberkochen, Germany) and evaluated using the software package ZEN 3.1 (Carl Zeiss, Oberkochen, Germany). For the evaluation of nuclear DNA repair proteins, 10 × 10^3^ cells were plated on glass cover slides coated with collagen-type I in a 24-well plate. After incubation in moderate hypoxia (1% O_2_) for 48 h and irradiation with 5 Gray (Gy) the samples were collected during the following 24 h at the given time points and stained following the same protocol as above with the appropriate primary antibody.

### 2.8. Apoptosis Assays

For quantification of apoptosis, caspase-3 activity was measured as described previously [29]. In brief, the cells were lysed in caspase-3-lysis buffer (50 mM Tris (pH 7.3), 150 mM NaCl, 1% (*v*/*v*) Nonidet P40). The protein concentration was calculated using a BCA Kit (ThermoFisher Scientific). Ten micrograms protein was prepared with 66 μM acetyl-Asp-Glu-Val-Asp7-amido-4-methylcoumarin (Ac-DEVD-AMC; A1086, Sigma) and 10 mM DTT in caspase-3 substrate buffer (20 mM HEPES (pH 7.3), 100 mM NaCl, 10% (*w*/*v*) saccharose (*w*/*v*), 0.1% (*w*/*v*) CHAPS), and incubated at 37 °C. Quantification was performed by measuring AMC fluorescence in black bottom 96-well plates at 430 nm every 10 min for 4 h. The data are displayed in bar graphs where the bars represent single time point measurements in the linear range of the reaction. As a second and independent apoptosis assay, PARP cleavage was assessed by Western blotting.

### 2.9. Neutral Comet Assay

Cells were seeded at a density of 1 × 10^5^ cells/cm^2^ and allowed to grow under normoxia or hypoxia for 24 h. Following irradiation, cells were trypsinized and embedded in 1% low-melting agarose. Cold lysis was performed according to the Olive protocol [30] with the lysis buffer containing 2% sodium lauryl sulfate, 0.5 mM Na_2_EDTA, 0.5 mg/mL proteinase K, pH 8, stored at 4 °C in the dark overnight, and incubated another 24 h in high salt buffer (1.85 M NaCl, 0.15 M KCl, 5 mM MgCl2, 2 mM EDTA, 4 mM Tris, 0.5% Triton X-100, pH 7.5) at 4 °C. Slides were then incubated in cold electrophoresis buffer (90 mM Tris buffer, 90 mM borat, 2 mM Na_2_EDTA, pH 8.5) for 20 min at room temperature. Electrophoresis was run at 20 V, 300 mA for 25 min, followed by staining with SYBR gold (Invitrogen). Comets were analyzed using the open-source image analysis software OpenComet v1.3.1. Fifty cells were counted per slide. Two slides were counted per condition for each experiment. 

### 2.10. Statistical Analysis

All assays were performed at least three times and data are expressed as means ± standard deviation. Statistical analysis was performed with GraphPad Prism 6 using two-way ANOVA or Student’s *t*-test for comparing two groups. Bonferroni post hoc tests were applied where indicated. Significance is presented as * *p* < 0.05, ** *p* < 0.01, *** *p* < 0.001.

## 3. Results

### 3.1. Generation of a Mouse Model with Tamoxifen-Inducible HIF-1α Knockout

Given that a global knockout of HIF-1α leads to embryonic death [31,32], we used Cre-Lox technology to generate mice with a permanent, inducible HIF-1α KO in adult life. Specifically, we crossed C57/BL6 mice with a conditional KO of HIF-1α [33] with ROSA26 CreERT^ki/ki^ mice. The Hif1a^fl/fl^ mice carried loxP sites flanking Hif1a exon2 (E2) which is essential for DNA binding and dimerization with ARNT following transcriptional activation. Cre-ER^ki/ki^ mice carry the knock-in (ki) mutation of the Cre recombinase fused to the mutated estrogen receptor, which can be activated by the estrogen receptor ligand tamoxifen. Therefore, tamoxifen application induced E2 excision, and thus prevented functional activation of HIF-1α. (Figure 1)

### 3.2. Characterization of mHDC Morphology and Function

In order to develop an in vitro system of functional murine hepatocytes, cultured cells should accurately reflect both function and morphology as observed in vivo. When transferred to our laboratory, the hepatocytes displayed a regular, flat, and polygonal morphology with rounded nuclei and a granular cytoplasm (Figure 2A). To determine whether the cell lines had preserved hepatocyte function under in vitro conditions, we investigated the expression of hepatocyte specific genes which are considered as hallmarks of the hepatocyte phenotype. Functionally, adult hepatocytes secrete a variety of serum proteins such as albumin and acetylcholine esterase. All cell lines produced albumin (Figure 2B) as demonstrated by Western blotting. Serum acetylcholine esterase is responsible for the hydrolysis of the neurotransmitter acetylcholine in blood. The enzyme exists in multiple molecular forms and is produced mHDC by enzyme linked immunosorbent assay (ELISA). The results pointed out that all cell lines secreted acetylcholine esterase (Figure 2C) into the cell culture media. The differences in secretion most likely reflect variability in cell density and were neither caused by tamoxifen nor by hypoxia. As demonstrated by immunohistochemistry, all isolated cells were positive for the hepatocyte markers cytokeratin 8 and 18 antibodies. (Figure 2D). As a fundamental marker of uncompromised cell metabolism, we assessed proliferation by cell counting (Figure 2E,F). We observed a minor reduction in cell numbers in HIF-1α KO mHDC which did not reach statistical significance. 

### 3.3. HIF-1α Knockout Promotes Cell Death and Increases Radiation Sensitivity

The efficiency of the inducible HIF-1α KO on the protein level was analyzed by Western blot (Figure 3A). Cre-positive cells expressed a shorter version of HIF-1α under hypoxic conditions (Hx) which lacked the DNA binding domain and was thus inactivated. Full length HIF-1α was not detectable in the tamoxifen treated cells. All mHDC expressed HIF-2α with a normal induction pattern when comparing normoxia and hypoxia (Figure 3B). In addition, the mRNA levels of HIF-1α and its specific targets BNIP3, PFK, NOS3, and GLUT-1 were quantified by qPCR. Quantification of cDNA generated from mHDC demonstrated that the mRNA levels of all four target genes were downregulated and no longer hypoxia-inducible in the KO cells (Figure 4). Next, we tested whether HIF-1 regulation was still functional after IR in wildtype and HIF-1α deficient cells (Figure 5A). We observed that HIF-1α was induced in all wildtype cells, while the shorter version of HIF-1α which we had detected in Figure 3A was barely detectable. The expression of HIF-2α appeared to be slightly reduced in hypoxic Tam treated HIF-1α knockout mHDC before (Figure 3B) and after irradiation (Figure 5B). We then determined by MTT assay whether depletion of HIF-1α had any effect on cell viability in mHDC before and after treatment with IR. We observed a HIF-1 independent reduction in viability in response to tamoxifen treatment. In addition, viability was significantly reduced in hypoxic knockout mHDC. The most striking and highly significant decrease was demonstrated in hypoxic HIF-1α-depleted cells 48 h after radiation treatment, which is in line with a protective effect of HIF-1 in irradiated cells (Figure 6). Furthermore, we analyzed the impact of HIF-1α-depletion on clonogenic survival following irradiation. Clonogenic survival assays revealed that under normoxic conditions no differences between wild type and knockout cells were detectable, while under hypoxic conditions, cells with a HIF-1α KO showed a significant decrease in mean survival fraction 10 days after seeding (Figure 7). 

### 3.4. HIF-1α Knockout Modulates Apoptosis in Response to Radiation

In previous studies, HIF-1α was described as a mediator of apoptosis induction in cancer cells [34]. Thus, in this study the effect of HIF-1α on radiosensitivity was also investigated by analysis of apoptosis. To analyze HIF-1α effects, the activity of pro-apoptotic caspase-3 was assessed. A significant increase in apoptosis in HIF-1α KO cells as compared to control cells after IR (Figure 8A) indicated that HIF-1 is a protective factor in this setting. To corroborate these results, the HIF-1α-depleted mHDC cells were irradiated and subjected to Western blotting using a poly [ADPribose] polymerase 1 (PARP-1) antibody which detects endogenous levels of full-length PARP-1 as well as cleaved PARP-1 as an indicator of apoptosis. The mHDC were irradiated with 5 Gy and the HIF-1α KO and control cells were lysed after 1, 4, 8, or 24 h in hypoxia. The experiments demonstrated an increased PARP cleavage in HIF-1α-depleted cells within the first 8 h after IR thus confirming the caspase-3 results (Figure 8B). 

### 3.5. HIF-1α Deficiency Leads to Increased DNA Damage and Delay in DNA Repair

The working mechanism of the induction of DNA damage by IR is through the generation of double-stand breaks (DSB). The DNA-damage response (DDR) and the repair ability of the cells is known to be altered in hypoxia which leads to genomic instability and, as a consequence resistance to radiation therapy [17]. HIF-1α was described to interact with crucial factors of DDR and checkpoint control [35]. Therefore, we investigated the effect of HIF-1α KO on IR-induced DNA damage by analysis of DSB generation using γH2AX immunohistochemistry. The γH2AX form of H2AX is formed by phosphorylation on the site of DSBs in the nucleus. Following irradiation with 5 Gy, staining of the nuclei revealed a highly significant (*p* < 0.001) difference in the generation of the γH2AX foci and a delay in removal when comparing HIF-1α-depleted and HIF-1α-containing cells during a period of 24 h (Figure 9A–C). In addition, we tested whether the addition of dimethyloxalylglycine (DMOG), a compound widely used for the oxygen-independent induction of HIF, recapitulates the effects generated with the hypoxic induction of HIF in mHDC. DMOG did not induce γH2AX foci in the absence of IR. Remarkably, however, in normoxia, DMOG-treated HIF-1α knockout cells developed more γH2AX foci after irradiation than all other cells. As the only genetic difference between 22 KO Tam(+) and 22 KO Tam(−), mHDC is HIF-1α deficient, this result demonstrated that HIF-1 promoted resistance to IR in mHDC. On the other hand, this experiment also showed that DMOG increased damage induced by IR in normoxia in all HIF-1 wildtype cells while it reduced damage in the same cells in hypoxia. As a second and independent DNA damage assay, we tested whether the same effect was detectable using the neutral comet assay, a method used to detect DNA damage, including DSBs, in individual cells [36]. The tail moment (percentage of DNA in the tail) was 1.6–1.9 fold increased, 5 and 30 min after IR by HIF-1α depletion under hypoxic conditions (Figure 10A,C), consistent with an increased number of DSBs after IR, while under normoxic conditions most DSBs were repaired 30 to 60 min after IR (Figure 10B), probably via NHEJ, which is the fastest DSB repair pathway. 

## 4. Discussion

Cell cultures of differentiated primary hepatocytes of human or rodent origin would be of enormous value in biomedical research. Unfortunately, the establishment of such cells in vitro has been hampered by rapid and inevitable de-differentiation in a cell culture environment. The earliest attempt to solve this problem was the use of cells taken into culture from hepatocellular carcinoma, such as the cell lines HepG2 and Hep3b, which have been commercially available for almost forty years now. However, these cells carry the inherent uncertainty as to whether they truly represent differentiated hepatocytes. For example, HepG2 express α-fetoprotein [37], a marker of fetal liver cells or HCC. The same type of problem accompanies the use of more recently developed cell lines, such as HepaRG and induced pluripotent stem cells (iPSCs), differentiated into hepatocytes, as expertly reviewed very recently [38]. Recently, another option became available: the use of “upcyte^®^ cells” [27,39] initially intended to study hepatotoxicity and human xenobiotic metabolism. Within this cell culture protocol, differentiated human hepatocytes from adult donors are isolated from liver biopsies and transduced with proliferation inducing genes. The resulting cells are not fully immortalized but grow up to 45 population doublings, depending on donor and species. Importantly, proliferation can be stopped at any time by the withdrawal of growth factors. Therefore, this system combines the generation of large cell quantities with the use of differentiated, quiescent hepatocytes in the actual experiments. In a gene expression analysis, upcyte cells more closely resembled primary human hepatocytes than HepG2 cells [40]. In our study, we have, for the first time, produced upcyte cells from murine liver. This procedure allowed the generation of differentiated hepatocyte cultures from wild type but also from genetically modified mice. The cells grew to confluence in a monolayer, which was important because our study required uniform oxygen delivery to all cells. With respect to the expression of hepatocyte differentiation markers such as cytokeratin 8/18, albumin, and acetylcholine esterase, our cultures remained stable over at least approximately 2 months. 

In our project, we used mice carrying inducible global Cre expression and biallelic loxP flanking of exon 2 of the transcription factor subunit HIF-1α. Thus, induction of Cre expression by tamoxifen treatment led to the irreversible destruction of the Hif1a gene in the cell cultures. Importantly, use of the upcyte protocol allowed the generation of all relevant control cultures: cells isolated from Cre negative mice were used to control for Cre-independent tamoxifen effects and cultures from Cre positive animals that were not treated with tamoxifen allowed us to compare uninduced versus induced state in syngeneic cells. As even the cells of a distinct culture are not clonal, it was important to demonstrate functional inactivation of HIF-1α by a complete lack of target gene induction in hypoxia by qPCR. Vice versa, the cell culture of HIF-1α negative mHDC, months without apparent growth delay as compared to control cells confirmed that HIF-1 is not a factor of vital importance as long as the cells are sufficiently supplied with oxygen.

HIF-1 has a number of adaptive effects on cell metabolism most prominently in the event of a lack of oxygen supply, i.e., hypoxia [41]. Regulation of HIF mostly occurs on a posttranslational level: at high levels of oxygen, the α-subunit is hydroxylated and degraded by the ubiquitin proteasome pathway [42]. Additional regulatory pathways, such as the activation of the transcription of the Hif1a gene and phosphorylation events, have also been reported [43]. In hypoxia, degradation is blocked because hydroxylation requires molecular oxygen, and HIF-1α translocates to the nucleus to induce target gene transcription following dimerization with its β-subunit [44]. 

Remarkably, the role of HIF-1 in hepatocytic stress responses has remained controversial over the last years. Therefore, our study was focused on the role of HIF-1 in the induction of proliferation and in apoptosis induction in response to IR. We chose IR because it represents a reproducible and quantitatively defined way to induce cellular stress, which is applicable in normoxia as well as in hypoxia. In addition, studying IR in liver cells holds a translational aspect because IR is presently regarded as an increasingly important therapeutic option in advanced hepatocellular carcinoma. Technical advances such as stereotactic extracorporal radiation have led to an improved outcome when combined with local interventions [45]. Thus, radiation responses of more or less differentiated hepatocellular carcinoma cells but also of untransformed hepatocytes in the vicinity of the tumor need to be investigated. Many types of cancer exhibit radiation resistance when hypoxic cells activate the HIF pathway. High levels of HIF-1α correlate with tumor progression and poor patient outcome [46]. Having established permanently HIF-1α-deficient mHDC cultures, we regarded our cells as a perfect tool to investigate HIF effects on proliferation, induction of apoptosis, and sensitivity to ionizing radiation. Reassuringly, we did not observe substantial knockout effects of HIF-1α in normoxia. When we assessed the proliferative activity by counting the cells in normoxia and in hypoxia, we observed an insignificant reduction in growth in hypoxic knockout cells. Hypoxia plus HIF-1α KO did not stimulate apoptosis induction as compared to hypoxia alone in the absence of IR. Interestingly, we also observed a reduction in viability as assessed by MTT assays in response to tamoxifen treatment. In essence, we interpret the reduction in viability in hypoxic HIF-1α knockout cells as a combination of two mildly inhibitory effects on proliferative activity: tamoxifen and the HIF-1α KO. A reduction in growth may be caused by a disturbance in energy metabolism, which has been reported for other cell lines previously [47]. However, in response to IR, our results showed that HIF-1α depletion led to elevated levels of apoptosis in mHDC, as shown by caspase activity assays and PARP Western blotting. Based on colony formation assays, long-term survival of hypoxic mHDC cells with HIF-1α KO was impaired after IR with 6 Gy. Therefore, our data suggest that the deletion of HIF-1α leads to an increase in apoptosis and a reduction in clonogenic potential, which demonstrates an increase in radiosensitivity. Previously it has been reported that HIF-1α knockdown leads to an impaired expression of survivin, but to elevated expression of caspase-3 and Bax proteins [48], which may contribute to elevated apoptosis levels and increased radiation sensitivity, as demonstrated in our mHDC.

Previous studies have found apparently conflicting results with respect to the interaction between HIF-1α and DNA damage repair. We therefore focused on the role of HIF-1 KO in DDR in response to IR. We first analyzed the induction of γH2AX foci by immunohistochemistry. The protein γH2AX is a phosphorylated form of the histone H2AX that is generated at sites of DSBs during the initiation of DDR. Detection of γH2AX foci therefore points to cellular recognition of DNA damage rather than commitment to a distinct pathway of DNA repair. Our experiments demonstrated that γH2AX foci are increased in HIF-1α KO cells in comparison to wild type cells after IR in hypoxia. Of note, we were able to recapitulate the damage promoting effect of loss of HIF-1 in HIF-1α knockout cells in normoxia by the addition of DMOG, which inhibits prolyl hydroxylases that initiate oxygen-dependent degradation of HIF-α. In this experiment, cells that were unable to induce HIF-1 because of the knockout developed more γH2AX foci than wildtype cells which reinforces our interpretation that HIF-1protects the cells from IR-induced DNA damage. Surprisingly, DMOG treatment of wildtype cells led to increased DNA damage in normoxia but reduction in damage in the same cells in hypoxia. While the hypoxic DMOG effect in wildtype cells is compatible with protection by HIF-1 activation, the normoxic effect clearly is not. It should be considered that DMOG is a pan-hydroxylase inhibitor, which is probably active on more than 30 2-oxoglutarate-dependent dioxygenases with an even larger number of substrates [49]. Overall, our results indicated that HIF-1α depletion in mHDC leads to an increase in DNA damage after IR. This increase in damage and the previously discussed increase in apoptosis in the first hours after IR pointed to a possible alteration to DNA repair after HIF-1α depletion. Of note, the amount of DNA damage decreased after the first 6 h, indicating that surviving cells were, in principle, able to complete DNA repair. The DNA repair in the first hours after IR was mainly affected by a fast repair mechanism termed NHEJ. A study with murine fibroblasts reported that HIF-1α KO leads to an increase in γH2AX [50], which is in line with our results. In addition, a number of studies report an increase in NHEJ under hypoxia as a result of HIF-1 activation in glioblastoma [51] and squamous cell carcinoma [52], potentially due to HIF-1α-mediated upregulation of Ku70 and DNA-PK. In the light of these studies, it is plausible that the elimination of functional HIF-1 led to an increased amount of DNA damage which was caused by the downregulation of NHEJ in our mHDC. As a note of caution, it has been published that γH2AX foci may also be detectable in a HIF-1-dependent but DNA damage independent manner [53]. To exclude misinterpretation of our γH2AX data, we performed neutral comet assays, which detect DSBs more directly as a result of the enhanced mobility of the smaller DNA fragments. These assays confirmed that more DNA damage was detectable in the first hour after irradiation and that repair was completed but with slower kinetics, which corroborated the γH2AX results. A second important repair mechanism, homologous recombination repair (HRR) is more active in S and G2 phase of the cell cycle [54]. Alterations of HRR were also reported in response to hypoxia [55] but as our mHDC had been transferred to a quiescent state by the withdrawal of growth factors, we anticipate that hypoxia or HIF-dependent effects on HRR contributed to a lesser extent to our results, if at all.

In summary, we have established murine hepatocyte derived cells as a culture model for differentiated adult mouse hepatocytes using the “upcyte” technology. We observed that a permanent and complete KO of functional HIF-1α caused a high correlation between the decrease in cell survival and increase in DNA damage in response to ionizing radiation, specifically under hypoxic conditions. Our mHDC will be a highly useful tool in a more detailed analysis of HIF-1 effects on DNA repair pathways. In addition, they will also allow a far more general analysis of HIF-1 effects on hepatocyte cell cycle, metabolism, and responses to stress. 

## Figures and Tables

**Figure 1 cells-11-01671-f001:**
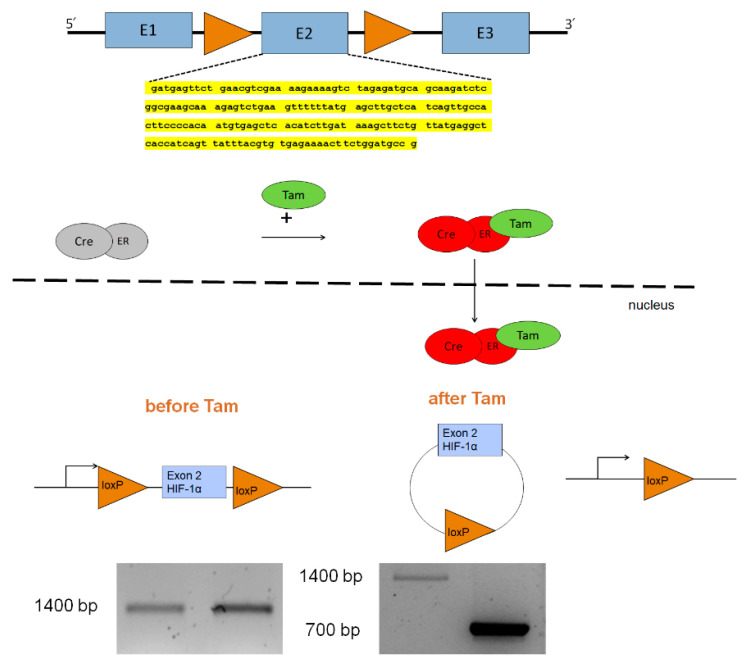
Strategy for generation of a conditional knockout of Hif1a. Mice were used which carry loxP sites flanking exon 2 (E2) of Hif1a [26]. Recombination was accomplished by tamoxifen induced expression of Cre from a ROSA26_CreERT2 locus. Loss of exon 2 resulted in the inability of HIF-1α to bind to DNA and HIF-1β. PCR genotyping demonstrated a shortened amplification product in the Hif1a^fl/fl^ Cre-ER^ki/wt^(Cre(ki)) mice after Tam treatment compared to the Hif1a^fl/fl^ Cre-ER^wt/wt^ (Cre(-)) mice, indicating the excision of E2. Yellow highlight: Nucleotide sequence of exon 2 of HIF-1α (GenBank AH006789.2).

**Figure 2 cells-11-01671-f002:**
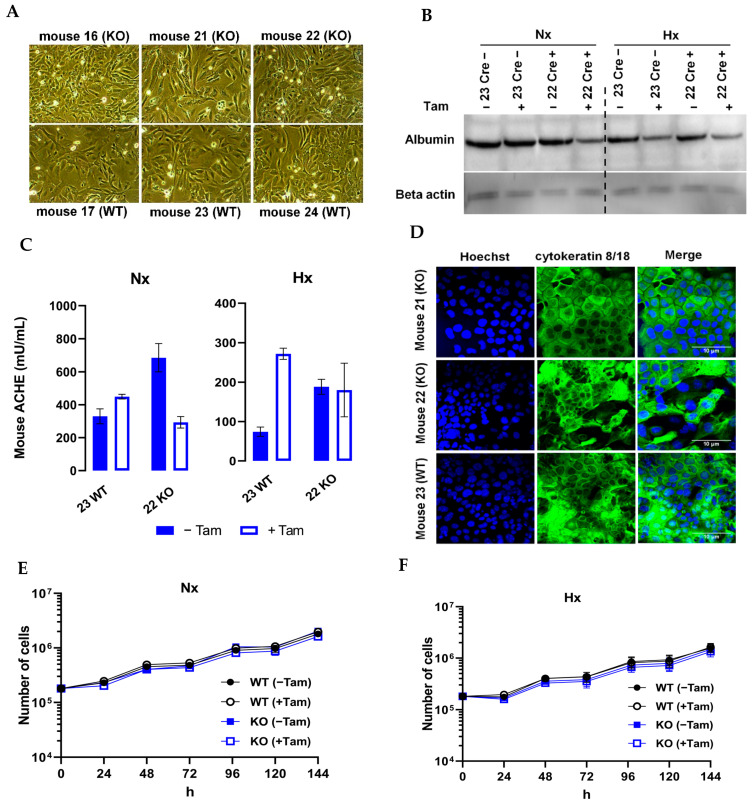
Analysis of differentiation of mHDC. Tam+ indicated tamoxifen treatment. NX designated the incubation in normoxia, HX in hypoxia (1% oxygen). (**A**) Phase contrast microscopy of nearly confluent cultures of mHDC monolayers. (**B**) Albumin Western blot of mHDC protein lysates. (Thirty micrograms per lane was loaded onto the gel. Hypoxic incubation was for 24 h). Cre-mHDC were isolated from Cre-negative mice, Cre+ were from Cre positive animals. Beta-actin served as a control for equal loading and protein transfer. (**C**) Cell culture media of mHDC were analyzed for secretion of murine acetylcholine esterase by ELISA. (**D**) Expression of CK8/CK18 was analyzed by immunofluorescence staining. Cell nuclei were counterstained with Hoechst33342. (**E**,**F**) Cell counting of mHDC under normoxic or hypoxic conditions. The mHDC were transferred to normoxia (NX) or hypoxia (HX) 24 h after seeding and counted for up to 5 days after transfer.

**Figure 3 cells-11-01671-f003:**
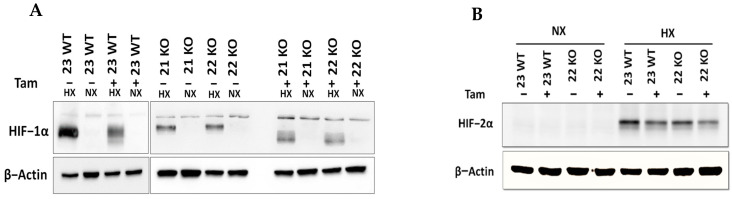
HIF-1α (**A**) and HIF-2α (**B**) protein expression in mHDC, as determined by Western blotting. HIF-1α KO was induced with tamoxifen (Tam). Cells designated “23 WT” were Cre negative, thus Tam did not induce a HIF-1α KO. The Cre positive cell lines “21 KO” and “22 KO” expressed a shorter non-functional version of HIF-1α after Tam treatment. The cells were incubated under normoxic (NX) or moderately hypoxic conditions (HX, 1% O_2_) for 4 h. Beta actin was used to demonstrate equal loading and transfer.

**Figure 4 cells-11-01671-f004:**
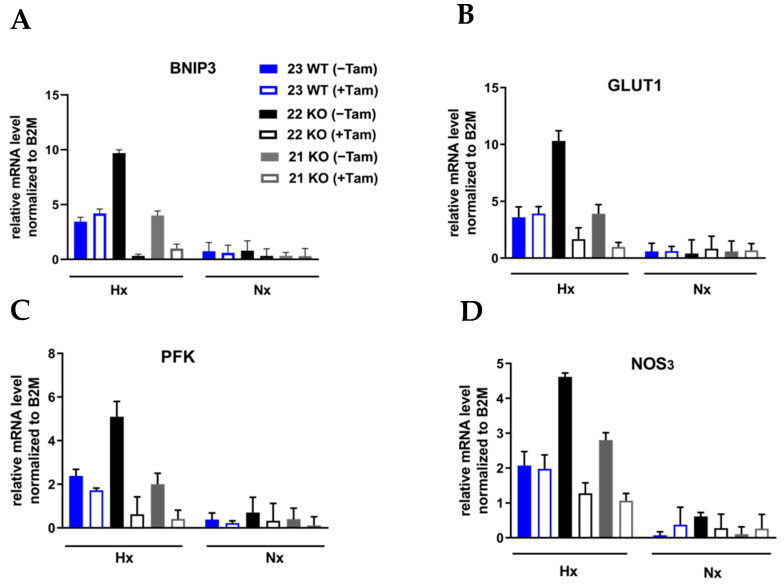
Analysis of mRNA levels of HIF-1α target genes in mHDC by RT-qPCR. The cells were grown overnight in normoxia (NX) or hypoxia (HX, 1% O_2_). The cDNA levels of the target genes (**A**) BNIP3, (**B**)GLUT-1, (**C**) PFK, and (**D**) NOS3 were analyzed by qPCR and normalized to the cDNA levels of the control gene ß2-microglobulin (B2M).

**Figure 5 cells-11-01671-f005:**
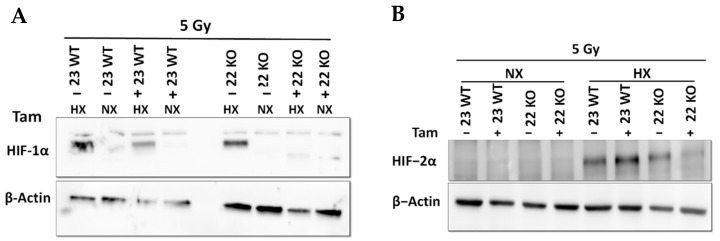
HIF-1α (**A**) and HIF-2α (**B**) Western blotting after 5 Gy irradiation. NX indicates normoxia, HX stands for hypoxia (1% O_2_) for 4 h.

**Figure 6 cells-11-01671-f006:**
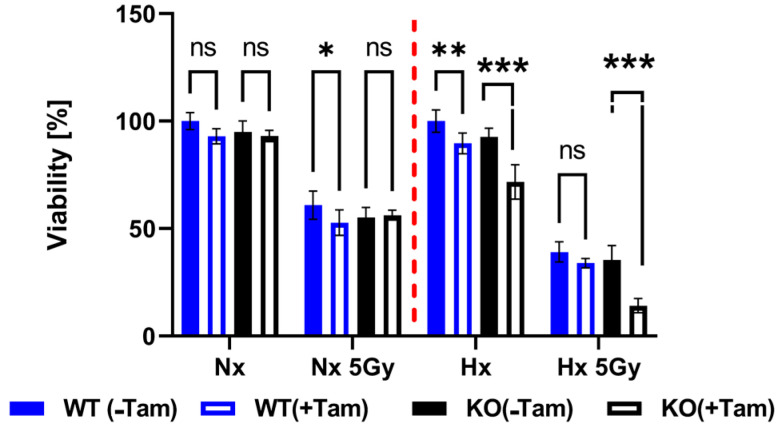
Quantification of cell viability as analyzed by MTT assays. The cells were cultured for 48 h in normoxia (NX) or in hypoxia (HX) and then irradiated with 5 Gy. Control cells were treated the same way but not irradiated. After IR the cells were transferred back either to normoxia or hypoxia. Forty-eight h after irradiation the cells were incubated with MTT and lysed 4 h later. Absorbance was measured at 540 nm. The values determined for Cre negative cells not treated with Tam were taken as reference values set to 100%. *n* = 10 samples per group were compared using two-way ANOVA (* *p* < 0.05; ** *p* < 0.01; *** *p* < 0.001; ns *p* > 0.05).

**Figure 7 cells-11-01671-f007:**
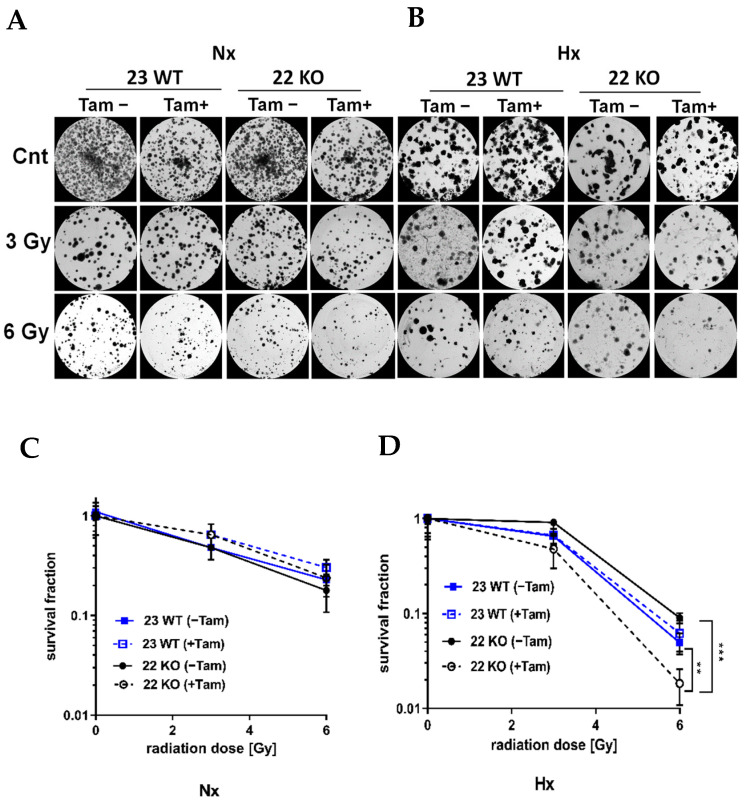
Survival of mHDC in response to irradiation with up to 6 Gy in normoxia (NX) or hypoxia (HX) as determined by colony formation assay (CFA). Hypoxic cells were cultured at 1% O_2_ in rat collagen coated 6-well dishes. After IR, the cells designated HX were placed in hypoxia for 10 days. The cultures were fixed, stained with Coomassie and colonies of at least 50 cells were counted. Photomicrographs in (**A**,**B**) show representative pictures of colony formation upon treatment with IR without or with Tam treatment for Nx (**A**) or moderately hypoxic conditions (**B**). Survival curves shown in (**C**,**D**) depict quantification of colony formation upon treatment in normoxia (**C**) and hypoxia (**D**). All groups consisted of *n* = 6 independent cultures. Groups were compared using two-way ANOVA. (** *p* < 0.01; *** *p* < 0.001).

**Figure 8 cells-11-01671-f008:**
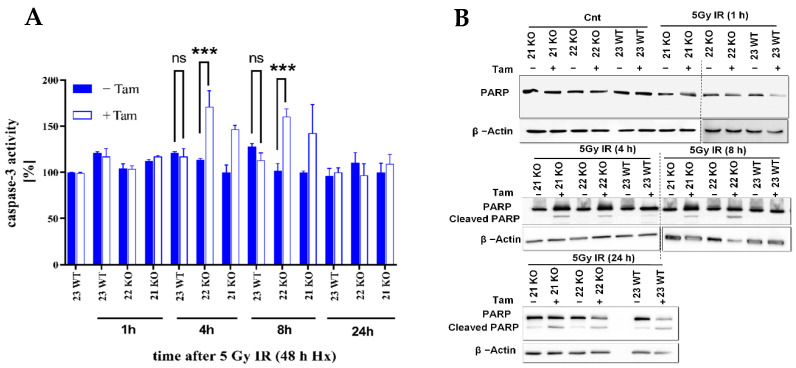
Analysis of apoptosis induction in hypoxic HIF-1α deficient mHDC. (**A**) The activity of caspase-3 was measured in mHDC lysates. Cells were irradiated and incubated in hypoxia for the time indicated. The cells were then lysed, and 10 µg protein were incubated with Ac-DEVD-amido-4-methylcoumarine. Generation of the fluorescent product was monitored at 430 nm. Caspase-3 activity data showed representative experiments with *n* = 6, each experiment was repeated three times. Graphs show mean ± SD, groups were compared using two-way ANOVA, *** *p* < 0.001, ns *p* > 0.05. (**B**) Western blot with a PARP-1 antibody which detects full-length PARP and a cleaved species with a lower molecular weight indicative of PARP cleavage, a signature of apoptosis. The cells were incubated in hypoxia for 48 h, then irradiated with 5 Gy, and transferred back to hypoxia for the time indicated. The cells were then lysed and subjected to SDS-PAGE and Western blotting. The data show a representative Western blot from three independent experiments.

**Figure 9 cells-11-01671-f009:**
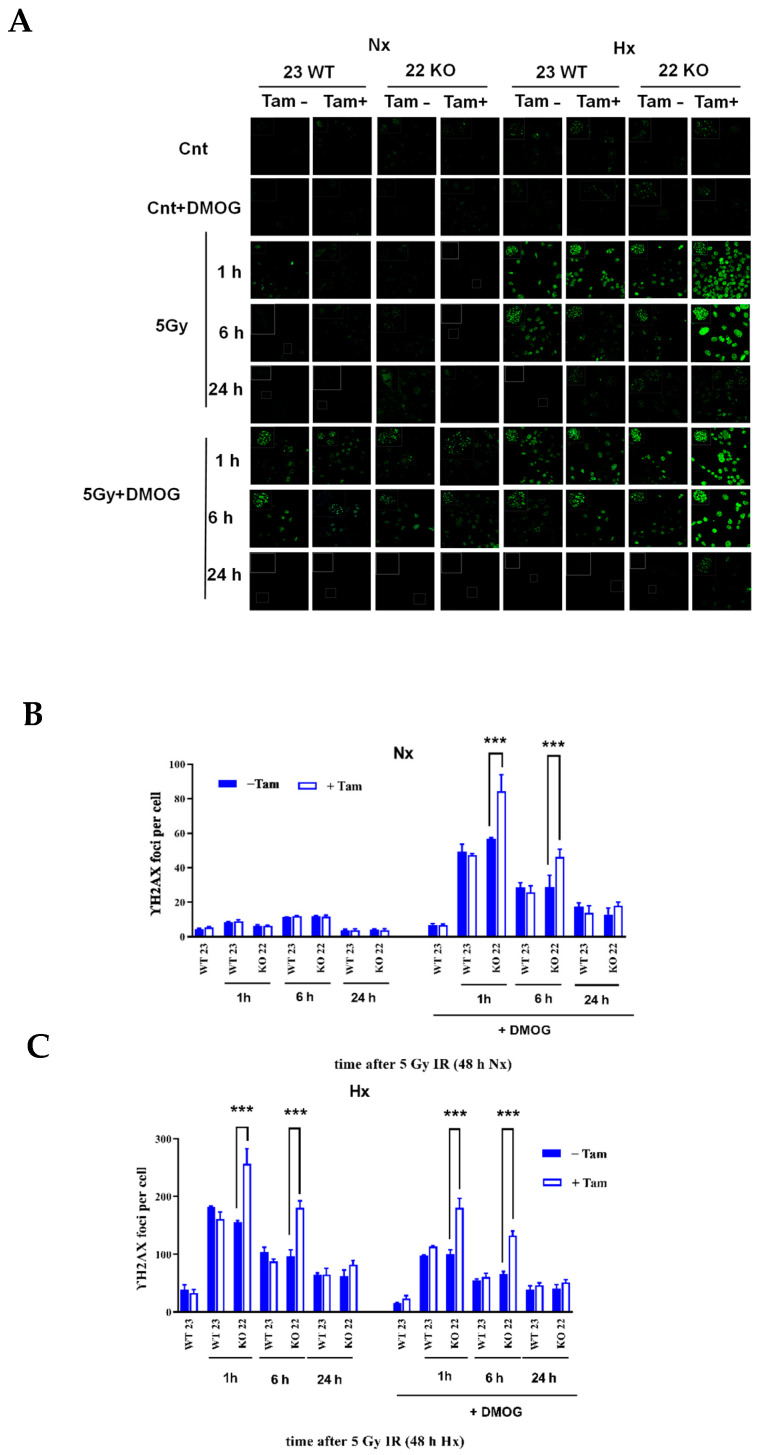
Formation of γH2AX foci after IR in HIF-1α knockout mHDC under normoxia or moderate hypoxia, and after treatment with the hydroxylase inhibitor DMOG. The mHDC were incubated in normoxia (NX) or moderate hypoxia (HX), respectively, for 48 h and then irradiated with 5 Gy. The cell cultures were then transferred back to hypoxia or normoxia and stained after the time indicated. Images were taken with a Zeiss LSM510 confocal microscope (**A**). Fields of vision with more than 50 nuclei were evaluated by counting the γH2AX foci in normoxia (**B**) or hypoxia (**C**). The first pairs of bars designated WT23 represent mHDC which contain HIF-1α and were not irradiated. Each experiment was performed three times. Graphs show mean ± SD, groups were compared using two-way ANOVA and the Bonferroni post hoc test, *** *p* < 0.001.

**Figure 10 cells-11-01671-f010:**
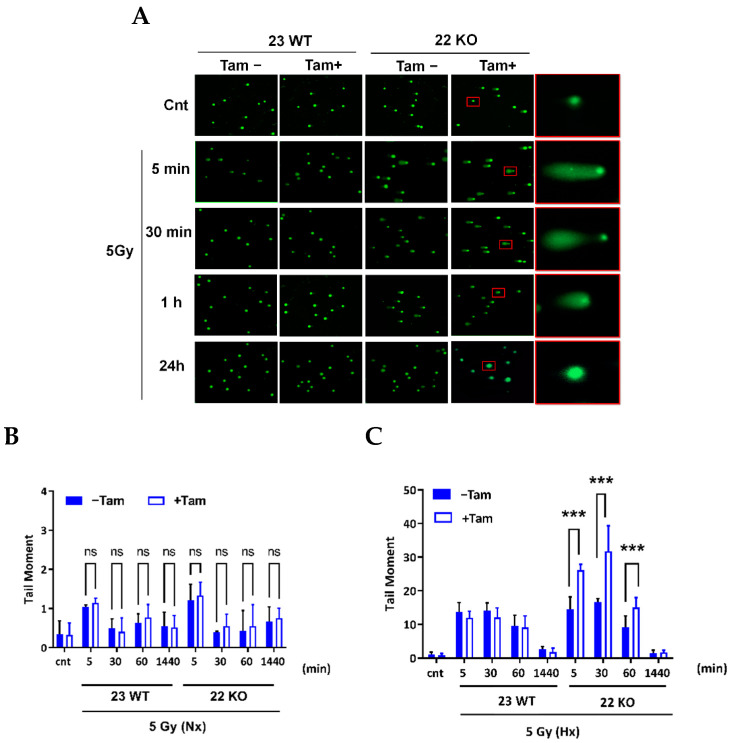
Effect of HIF-1α knockout on DSB repair mechanisms as assessed by a neutral comet assay. After IR, the cells were embedded in 1% agarose gels, lysed, and subjected to electrophoresis at 25 V, 300 mA for 25 min. (**A**) Representative fluorescence microscopy images are shown, red squares show KO cells in hypoxia chosen for higher magnification. The tail moment (percentage of DNA in the tail) was determined in 50 cells per slide with the open-source analysis program OpenComet v1.3.1 in normoxia (NX). (**B**) and hypoxia (HX) (**C**). Bars show mean ± SD, the groups were compared using two-way ANOVA, *** *p* < 0.001. ns *p* > 0.05.

## Data Availability

Experimental data and protocols are available from the authors upon request. Both, *Gt(ROSA)26Sor^tm9(Cre/ESR1)Arte^* x Hif1a mice and mHDC carrying the Cre/ESR1 knock-in, are covered by a license agreement with Taconic Biosciences and, therefore, cannot be handed over without Taconic’s prior consent. In addition, mHDC are covered by a Material Transfer Agreement with upcyte technologies and cannot be transferred without upcyte’s prior consent.

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
