# Peer review of "Depletion of HIF-1α by Inducible Cre/loxP Increases the Sensitivity of Cultured Murine Hepatocytes to Ionizing Radiation in Hypoxia"

_cells, 2022, doi:10.3390/cells11101671_

Round 1
Reviewer 1 Report
Hamidi et al have established longer standing cultures of murine hepatocyte derived cells (mHDC) as a novel in vitro model to analyse the role of HIF-1α in apoptosis induction, DNA damage repair and sensitivity to ionizing radiation (IR). In general, primary hepatocyte cultures are very difficult to maintain and the ability to establish longer standing cultures enables many further studies. It is also very interesting that the data suggest that HIF-1a behaves like a tumor suppressor. In general, the manuscript is very nicely put together and the data is solid. I only have a few minor comments.
What were the lentivirus-mediated gene transfers that induced cell proliferation?
Altogether 3 HIF-1a KO and 3 control lines were generated; however, in most figures only a few were analyzed. Did the lines within the genotype differ from each other? Fig. 6A would suggest some differences.
Were there any obvious differences in the metabolism between the HIF-1a WT and KO lines? E.g. in ATP production? Could anything be concluded from the MTT assay?
In Fig. 6A x-axis; alight the 1h, 4h, 8h and 24 (add h) correctly.
In Fig. 1; the panels partly overlap and mask each other.
Author Response
Reviewer #1
We are pleased to note the reviewer’s favourable comments. Our replies to the points raised are as follows:
- The reviewer requests information about the lentivirus-mediated gene transfers. The specific protocol which has been used by upcyte technologies on our mouse hepatocytes cannot be published as it is in commercial use. As mentioned in our manuscript, however, the upcyte procedure is based on a previous publication (Levy et al., Nat. Biotechnol. 2105, ref. [39] of our manuscript). Within this report, hepatocytes were tranduced with lentiviral particles inducing the human papilloma virus genes E6 and E7 which in turn induced expression of the oncostatin M receptor. We have modified the manuscript accordingly, please see section 2.1 of the revised manuscript.
- The reviewer remarks that not all hepatocyte lines that we have generated were used in each experiment. Indeed, we have generated three mHDC from Cre-negative mice and three mHDC from Cre-positive mice. We can confirm that we have not seen significant, reproducible differences between the genotypes. However, within these groups the cell cultures were not equally robust, i.e. sometimes fluctuations in proliferation and morphology were observed. According to upcte technologies this phenomenon also occurs in human upcyte hepatocytes and is most probably caused by random integration of the HPV genes. Therefore, for further experimentation we have chosen the lines which were easiest to handle.
- The reviewer asks whether we have observed differences in the metabolism between the HIF-1a WT and KO lines. We would regard growth as the most general parameter of undisturbed cell metabolism. Therefore, in response to this point, we have integrated the new Fig. 2 E and F into the manuscript which shows that proliferation as analysed by counting of the cells demonstrates that there is a trend towards reduced cell counts in KO cells which, however does not reach statistical significance. The MTT test shows reduction of viability in response to tamoxifen in all genotypes which probably adds to a minor reduction of cell numbers to give a seemingly significant result. The text has been modified accordingly in section 3.3 and the discussion.
- We would like to thank the reviewer for pointing this out. We have corrected the x-axis of the figure which is Fig. 8A of the revised manuscript.
- The figure (Fig. 2 of the revised manuscript) has been re-arranged to avoid overlap of the panels.
Reviewer 2 Report
In this study the authors aimed at establishing cultured murine hepatocyte derived cells (mHDC) as an in vitro model to analyse the role of HIF-1α in apoptosis induction, DNA damage repair and sensitivity to ionizing radiation (IR). They used Cre-Lox technology to generate mice with a permanent, inducible HIF-1α KO in adult life after tamoxifen treatment.
They found that in moderate hypoxia, HIF-1α deficiency increased IR-induced apoptosis and significantly reduced the surviving fraction of mHDC as compared to HIF-1α expressing cells in colony formation assays. Furthermore, HIF-1α knockout cells displayed increased IR-induced DNA damage as demonstrated by increased generation and persistence of γH2AX foci. HIF-1α deficient cells showed delayed DNA repair after IR in hypoxia in neutral comet assays which may indicate that non-homologous end joining (NHEJ) repair capacity was affected.
They conclude that the mHDC cells could be a highly useful tool for studying the hypoxia effects on DNA repair pathways, cell cycle, metabolism and responses to stress.
The study is interesting and well designed. The results are clear and well presented.
There are minor points that need to be addressed before publication.
In the 3.1 Results a scheme of HIF-1alpha knockout could be added.
Paragraph 3.2 looks like a material and methods paragraph rather than a results paragraph. Please change and/or add figures for this paragraph.
In the legend please explain better Figure 1 B : Tam? HX, NX? Cre-? Cre+? Time of hypoxia and O2 percentage? Also show a loading control for this figure.
In figure 2 and in all figure legends, please explain what HX and NX stand for (I suppose hypoxia and normoxia).
In figure 2 please explain how long was the hypoxic condition and if the protein lysate is total or only nuclear (HIF-1alpha accumulates in the nucleus).
Author Response
Reviewer #2
The authors would like to thank the reviewer for the positive and encouraging remarks. Our responses to the points of criticism are as follows:
- As suggested, we have added a scheme presenting the knockout strategy – please see the new Fig. 1 of the revised manuscript.
- We followed the reviewer’s advice merged this paragraph into Materials and Methods, section 2.1.
- We have explained the abbreviations used in Fig. 1B and explained the experimental conditions more comprehensively. Beta-actin is shown as a loading control.
- In Fig. 2, we have also clarified the abbreviations.
- We have given more details related to the experimental conditions and the preparation of the samples. Details which apply to all Western blot, e.g. use of whole cell lysates are mentioned in the Methods section now.
Reviewer 3 Report
In this manuscript Hamidi el al. report the inactivation of HIF1a in isolated hepatocytes from mice harboring a HIF1a loxP locus and expressing a tamoxifen-inducible Cre. Moreover they show that HIF1a-deficent hepatocytes are more vulnerable to ionizing radiation specifically in hypoxic conditions. Under these conditions, HIF1a-deficent hepatocytes show reduced survival as well as signs of increased apoptosis and DNA damage. The novelty of the study can be considered limited taking into consideration previous studies in other cellular models (PMID: 16517405, PMID: 16517406, PMID: 22100406). Independently of its novelty the following comments should be addressed.
Major comments:
1.- Authors should assess whether an initial HIF1a activation by PHD or VHL gene inactivation precondition WT hepatocytes to be protected against a subsequent ionizing radiation exposure. Ideally this experiment should be performed using mice harboring Vhl LoxP (or PHD-1,-2 or -3 LoxP) locus with the expression of the tamoxifen-inducible Cre. If authors do not have these mice available, they should assess whether an initial HIF1 activation by PHD inhibitors (like DMOG or others) protect hepatocytes against a later ionizing radiation in WT hepatocytes but not in the same extent in HIF1a-deficent hepatocytes.
2.- Are reactive oxygen species (ROS) involved in the increased vulnerability of HIF1a-deficent hepatocytes to ionizing radiation?. Authors should assess first whether HIF1a-deficent hepatocytes increases the levels of reactive oxygen species specifically in hypoxic conditions or hypoxic conditions plus ionizing radiation and secondly whether antioxidant agents prevent the increased vulnerability of HIF1a-deficent hepatocytes to ionizing radiation.
3.- Authors should assess HIF2 in their isolated hepatocytes. Do isolated hepatocytes express this other HIF isoform?. Authors should assess HIF1a expression not only in hypoxia alone (Figure 2) but also in hypoxia plus ionizing radiation conditions.
4.- If possible authors should try an in vivo model of hepatic ionizing radiation in their HIF1a-deficient mice, which are viable after HIF1a inactivation in adult life. This approach will help to conclude that HIF1a acts as a hepatocyte protector against ionizing radiation.
5.- Authors mention that “Cre expression and the consecutive gene KO were induced by adding 500 nM 4H-tamoxifen to the hepatocyte cultures for three days”. After these three days, 4H-tamoxifen is then washed before hypoxia and/or ionizing radiation exposure?
Author Response
Reviewer #3
Initially,the reviewer correctly sums up design and results of our study. Then the reviewer states that the novelty of our study is limited and mentions three studies published by others in 2006 and 2011. Interestingly, in none of these studies hepatocytes or even liver derived permanent cell lines were used, and in none of them irradiation or DNA damage were analysed. Instead, the authors of these studies, which are truly important and have moved the HIF field forward, investigated oxygen consumption and ROS generation in HIF deficient renal, lymphocytic and fibroblast cell lines. However, because none of these articles discloses results from our manuscript, we feel that the serious comment “lack of novelty“ cannot be based on these studies.
The reviewer goes on to suggest in vivo and in vitro experiments to further characterize the effects which we have observed. We have done our best to address these points experimentally within the time frame allowed by the journal to re-submit. Specifically, we comment on the points raised as follows:
- The reviewer suggests to analyse hepatocytes with long standing activation of HIF-1α due to inactivation of PHD or pVHL. Even if we had the animals at hand, it would take several months to receive animal experimentation permission and again months to breed and to generate the hepatocyte cultures from these mice. As an alternative, the reviewer suggests to use dimethyloxalylglycine (DMOG), a compound widely used to induce HIF by PHD and FIH inhibition. DMOG is a pan-hydroxylase inhibitor which is probably active on more than 30 2‑oxoglutarate dependent dioxygenases with an overall unknown number of substrates. Most remarkably, we observed that normoxic DMOG-treated HIF-1α knockout cells develop more γH2AX foci, i.e. more DNA damage, after irradiation than all other cells. As the only genetic difference between 22 KO Tam(+) and 22 KO Tam(-) mHDC is HIF-1α deficiency, this result reinforces our interpretation that HIF-1 promotes resistance to IR in mHDC. On the other hand, this experiment demonstrates that DMOG increases damage in normoxia in HIF-1 wildtype cells while it reduces damage in the same cells in hypoxia. The molecular basis of the latter effects is currently unclear, but obviously not related to HIF-1α. We have included this result as Fig. 9 in the revised manuscript.
- The reviewer asks whether HIF-1α deficiency affects ROS generation and whether ROS mediate the effects that we have observed. … (DELETED, E.M.). Therefore, and to keep the manuscript focused, we have decided not to include these results in the manuscript, but we are happy to provide the reviewer with the following preliminary figure:
(DELETED, E.M.)
- As recommended, we have included Western blots for HIF-1α before and after ionizing radiation and Western blots for HIF-2α which is expressed by all mHDC at a similar level – please see modified Fig. 3 and the new Fig. 5 of the revised manuscript.
- The reviewer requests analysis of an in vivo model of hepatic ionizing radiation. We agree that this model would be highly informative, but we are only aware of one group worldwide which has stereotactic mouse liver irradiation established (PMID: 27370150). We would like to respectfully point out that this approach would be beyond the scope of a revision for a cell biology journal.
- The reviewer asks for more precise information on the application of 4H-tamoxifen which we have provided in the revised version of our manuscript (please see paragraph 2.1).
Round 2
Reviewer 3 Report
The authors have addressed satisfactorily my comments. However authors should clarify the following
1.- Authors mention that “Next, we confirmed that HIF-1 regulation is still functional after IR in control cells (Fig. 5A), while expression of HIF-2α was not affected by IR (Fig. 5B). Data shown in figure 5 show that both HIF1 and HIF2 seem to be present in irradiated cells. Therefore it is not understood why authors include ‘while’ in this sentence. What are the differences between HIF1 and HIF2 regarding their expression after IR that authors refer to?
2.- Moreover authors should discuss why HIF2 expression is reduced in irradiated HIF1a-deficient hepatocytes.
3.- The new Figure 5 legend does not refer to A and B panels.
Author Response
The authors would like to thank the reviewer for the comments. Our responses are as follows:
1) "While“ is is indeed inadequate in this sentence. We have modified the paragraph as suggested.
2) We have mentioned that the level of HIF-2α in hypoxic HIF-1α deficient cells is reduced as compared to HIF-1α wildtype cells.
3) We have modified the legend to Fig. 5 as suggested.